# ADAPTIVE GENERATION OF PROGRAMMING PUZZLES

## ABSTRACT

AI today is far from being able to write complex programs. What type of problems would best teach computers programming, and how should such problems be generated? We suggest programming *puzzles* as a domain for teaching computers programming. A programming puzzle is a short program for a Boolean function $f(x)$ with the goal of finding an input that makes $f$ return `True`. Puzzles are *objective* in that one can easily test the correctness of a candidate solution, in contrast to other common specifications for program synthesis like input-output pairs or English problem descriptions. To address automatic puzzle generation, we propose a GAN-like algorithm called "Troublemaker" which can generate puzzles *adaptively* targeted at any given puzzle-solver. Rather than generating a single dataset of puzzles at random, it generates a diverse set of puzzles that are difficult for the solver. In our experiments, Troublemaker learns to generate challenging problems for a variety of state-of-the-art puzzle-solving techniques.

## 1 INTRODUCTION

Computers still perform worse than humans at elementary programming and mathematical reasoning problems, let alone the AI grand challenge of designing, analyzing, and implementing sophisticated algorithms. For example, many symbolic mathematical software packages fail to solve $n^{n^n} = 10^{10^{10}}$ for $n$. While programs can be taught to immediately answer such puzzles, it points to a fundamental lack of understanding that limits their ability to perform complex programming and mathematics.

The first question faced in the daunting task of teaching computers to program is, *how should one represent programming problems to teach computers?* The ideal representation may be different from what suits humans. Consider a question "What number follows the sequence 1-2-1-4-1-6-1?" (Barrett et al., 2018; Saxton et al., 2019). It requires: (a) understanding some English and (b) guessing which sequence the test-writer considers most natural among the many consistent ones. We refer to these biases as *human priors*. Program synthesis problems involve human priors as well. They are usually specified in English and/or by examples (Gulwani et al., 2012) such as *"Extract the area code from a phone number"* and/or "`(212)123-4567`" → "`212`". These involve prior knowledge about phone numbers, which may impede computers from learning the fundamentals of programming and math.[1]

In summary, to advance artificial reasoning, the ideal programming problem representation should be:

**Objective:** a candidate solution ought to be unambiguously validated as to whether or not it satisfies the specification. In particular, a problem-solving algorithm should be able to determine correctness without knowledge of English or consulting an answer key.

**Challenging:** Capture problems that human programmers can solve easily but which foil computers.

**Diverse:** Capture a rich range of useful programming problems from easy to hard.

**Unbiased:** Avoid dependence on human priors, such as language or spatio-temporal biases.

**Programming Puzzles**  As a better standard for evaluating and advancing artificial reasoning, we propose *Programming Puzzles*. In a puzzle, given the source code of a function $f$ in a fixed programming language (*e.g.* Python), the goal is to find an input $x$ such that $f(x)$ returns `True`.

---

[1]After creating NAPS (Zavershynskyi et al., 2018), a "Natural Programming Synthesis" dataset of programming contest problems with English descriptions and input-output examples, one of the authors later conceded that "ML, specifically natural language understanding, is not there yet." (Polosukhin, 2018)

```python
def f1(n: int, prefix=123456789):  #   find an integer n whose square begins with 123456789
    return str(n*n).startswith(str(prefix))

def f2(S: Set[int], n=11010010):  #      find a set S of powers of 10 summing to 11010010
    return sum({10**i for i in S}) == n

def f3(s: str):  #                        find a string s with 1,000 As but no two consecutive As
    return s.count("A")==1000 and s.count("AA")==0

def f4(x: List[Boolean]):  #                        solve a classic Boolean SAT CNF formula
    return (x[0] or x[1]) and (not x[1] or not x[2])

def f5(m: int, n=10987654321):  #            find a nontrivial integer factor of 10987654321
    return 1 < m < n and n % m == 0
```

Figure 1: Sample programming puzzles with valid answers $n = 111111111$ (Python: `int("1"*9)`), $S = \{1, 4, 6, 7\}$, $s =$ a concatenation of 1000 copies of "AB" (Python: `s="AB"*1000`), $x =$ [True, True, False], and $m = 7$. Problems range from easy to hard to unsolvable.

As Figure 1 illustrates, short puzzles can capture extremely difficult problems such as factoring or subset-sum as well as easy questions such as list reversal or solving $(x + 1)^{x+1} == 100^{100}$ for $x$. Importantly, such puzzles are objective – a candidate solution can easily be evaluated for correctness.

Not all programming problems can be nicely defined as puzzles. In some problems, writing the puzzle is as hard as solving the problem, *e.g.* long addition of two numbers. Other problems involve human priors, *e.g.* alphabetizing a list of names by last names, where language rules determine whether "`Mary De Leon`" is sorted under "`D`" or "`L`". However, puzzles with polynomial-time definitions constitute the complexity class $NP$, which contains $P$ problems such as shortest-path as well as hard problems such as factoring and finding a neural network with a sufficiently low training loss.

**Troublemaker** Our second main contribution is a "Troublemaker" algorithm that automatically generates puzzles in an *adaptive* manner to challenge any given puzzle-solving system. A primitive system may be given easy puzzles, while an advanced system may be given hard puzzles (or puzzles targeting a specific discovered weakness). Automatic problem generation has been applied to train program synthesis systems (Balog et al., 2016; Christakopoulou & Kalai, 2018) and in numerous related domains such as mathematical problems (Saxton et al., 2019) and IQ tests (Barrett et al., 2018). However, these problems were generated non-adaptively, often via random sampling. In this work, we use a host of state-of-the-art neural and symbolic machine intelligence techniques as target solvers, and show that Troublemaker learns to generate challenging puzzle datasets for all of them.

Troublemaker operates by *searching* for puzzles in a given grammar that would be hard for a given solver. The search process is made adaptive by guiding it with a trained neural network. The network is optimized to balance (a) difficulty of the puzzle w.r.t. the solver runtime, (b) diversity of generated puzzles, and (c) puzzle brevity. When the solver is trainable, this generation process results in an *adversarial* setup, where Troublemaker finds increasingly hard problems as the training progresses.

**Contributions** This paper introduces (1) programming *puzzles*, a class of objective problems that can be used to evaluate and improve the reasoning ability of existing AI systems; (2) an *adaptive* puzzle generator that can be targeted at any given puzzle solver(s). Note that we do *not* set out to create novel approaches for puzzle solving – we evaluate our puzzle generation on state-of-the-art neural and symbolic solvers. We focus on adaptive problem generation and establishing the puzzle domain as suitable for teaching machines how to program, leaving the improvements of puzzle solving techniques to future work.

## 2 RELATED WORK

**Program Synthesis** The prior work on program synthesis is too large to survey here (see the survey by Gulwani et al., 2017). As mentioned, prior work on program synthesis largely works from input-output examples $x_i, y_i$ where the goal is to find a "natural" function $f$ such that $f(x_i) = y_i$.

This representation is *subjective* – the issue is not that there are multiple correct programs but rather that the correctness of a given solution $f$ is debatable and cannot be determined from examples alone. Similarly, the research on *semantic parsing* (Liang, 2016) describes problems in natural language such as English, for which program correctness is even more debatable. From the point of view of expressing the intent of an end user, these representations may be quite natural and the ambiguity may be inherent to the problem. However, for the purpose of teaching computers programming and reasoning basics, such ambiguity impedes learning.

An exception is a closely-related work of Christakopoulou & Kalai (2018), which inspired this paper. They consider a specification that is itself a program that takes a whole problem-solving program as input. The specification program then generates a number of random inputs to the candidate program and ensures that the candidate satisfies them all. Similarly to our work, this specification in the form of source code is given to the problem solver.

A dual view of puzzles in that their solutions are programs, and the puzzle itself serves as a simple *formal specification*. Every puzzle solution can be then viewed as a constant program that satisfies this specification. In the programming languages community, the intent is often defined by a *formal specification* with the goal of finding any program that satisfies it (Manna & Waldinger, 1980; Bornholt & Torlak, 2017; Alur et al., 2018). The greater expressive power in some languages inherently means that it is not always trivial to determine whether a solution satisfies the specification.

**Evaluating Intelligence and Reasoning** Program synthesis systems that employ machine learning models are often trained on automatically generated programming problems (Balog et al., 2016; Christakopoulou & Kalai, 2018; Devlin et al., 2017; Parisotto et al., 2016). However, the way they generated problems was non-adaptive, *i.e.* independently of the solver. Recent work (Shin et al., 2019) has noted that such randomly generated datasets often fail to capture important properties of the desired program distribution, and thus result in a biased program synthesizer. Our Troublemaker framework allows adapting the dataset to the evaluation metric of a particular synthesizer.

As mentioned in Section 1, other tasks for evaluating and motivating progress in machine intelligence have been proposed, including mathematical word problems (Saxton et al., 2019) and IQ tests (Barrett et al., 2018). None of them achieve our desiderata of objectivity and independence of human priors. One notable example is the ARC task (Chollet, 2019), developed concurrently with this work. While it is also designed to establish a new standard in measuring intelligence and intentionally forgoes all human priors, its problem specification is still given by input-output pairs and thus is subjective.

**Adversarial and Curriculum Learning** The Troublemaker framework aims to generate short and diverse puzzles that are difficult for a given puzzle solver. As such, its main goal is to evaluate the solver and set an adaptive milestone to facilitate AI progress. However, a similar technique can be used to adaptively generate a training set, aiming to improve a given solver after discovering its weaknesses. In the machine learning community this is known as *curriculum learning* (Bengio et al., 2009; Graves et al., 2017; Sukhbaatar et al., 2018). The Alice-Bob framework of Sukhbaatar et al. (2018) is particularly relevant. In it, a *teacher agent* and a *student agent* engage in self-play where the teacher generates tasks for the student right at the edge of the student's "comfort zone", thus encouraging the student to continually improve. This is in contrast to Troublemaker, which over time generates problems *just beyond* the solver's comfort zone to expose its limitations. When a solver is trainable, Troublemaker proceeds by iterating between improving the solver and the puzzle generator.

Troublemaker, Alice-Bob, and similar iterative frameworks drive inspiration from *generative adversarial networks* (GANs) (Goodfellow et al., 2014). In addition to wide usage in improving generative models via learned adversarial objectives, the GAN ideas have also been recently used to improve robustness of trained agents (Pinto et al., 2017) or to automatically adapt their reward functions (Durugkar & Stone, 2018). These works are typically applied to reinforcement learning based environments, while Troublemaker focuses on the problem of generating hard and diverse puzzles without any assumptions on the target solver.

## 3 PROBLEM GENERATION AS A ZERO-SUM GAME

Our goal is to find a distribution over a diverse set of hard problems. The framework can be defined abstractly in terms of set $\mathcal{P}$ of problems and set $\mathcal{S}$ of solvers, where each problem $p \in \mathcal{P}$ computes a

Boolean function $p : \mathcal{X} \to \{\top, \bot\}$ for on some set $\mathcal{X}$, and each solver $S \in \mathcal{S}$ computes a function $S : \mathcal{P} \to \mathcal{X}$. Problem $p$ is said to be easy for $S$ if $p(S(p)) = \top$ and *hard* for $S$ otherwise. The set $\mathcal{S}$ captures the resource constraints on solvers (e.g., our experiments have a time budget $B$ by wrapping each solver in an appropriate timeout), and the set $\mathcal{P}$ captures restrictions on problems such as length limit and any other restrictions of interest such as solvability.

The Problem-generation game is a two-player zero-sum game between a *Generator* that chooses an arbitrary distribution $D \in \Delta(\mathcal{P})$ over $\mathcal{P}$, where $\Delta(T)$ denotes the set of probability distributions over set $T$, and a *Learning Solver* who chooses $S \in \mathcal{S}$. For example, we implement $\mathcal{S}$ as a set of neural guided solvers whose parameters are trained by the Learning Solver. The *payoff* to the Generator is $v(D, S) = \mathbb{E}_{p \sim D}[r(p, S, D)]$, where, $r$ is a *reward* function and $D(p)$ is the probability of generating problem $p$:

$$r(p, S, D) = \begin{cases} \lambda(-\log_2 D(p)) + (1 - \lambda) & \text{if } p(S(p)) \neq \top \\ 0 & \text{if } p(S(p)) = \top. \end{cases} \quad (1)$$

The reward is parameterized by $\lambda \in [0, 1]$ which offers a trade-off: at $\lambda = 0$, any hard problem earns the Generator a reward of 1, while at $\lambda = 1$ the reward is the expected negative log-likelihood over hard problems. The reward function may seem peculiar at first since it depends not only on problem's hardness but also on its probability of being generated. At $\lambda = 0$ the reward does not account for the diversity of the distribution of problems—against any solver $S$ a generator could maximize $v$ with the distribution that always outputs a single hard problem $p \in S$, yet we want $D$ to generate diverse hard problems. To account for diversity, the reward therefore depends on $D$ as well. As one varies $\lambda$ from 0 to 1, one expects the fraction of hard problems to decrease but the entropy of the hard problems to increase, a measure of greater diversity.

## 3.1 STATIC SOLVER

We consider two types of solvers. First, consider a fixed Solver $S$, i.e., the Learning Solver does not learn but instead just plays $S$, and let $H_S = \{p \in \mathcal{P} \mid p(S(p)) \neq \top\}$ denote the set of problems that are hard for $S$. It is not difficult to see that if the Generator only generates hard problems support$(D) \subseteq H_S$, then $v(D, S) = 1 - \lambda - \lambda H(D)$, where $H(D)$ is the *entropy* of distribution $D$. Entropy is a standard notion of diversity of a distribution, though other notions may be used.

**Lemma 1.** *For any $\lambda \in (0, 1]$ and any solver $S$ with $|H_S| \geq 3$ hard problems, the uniform distribution $\mathcal{U}_{H_S}$ over $H_S$ uniquely maximizes the Generator's payoff $v(\mathcal{U}_{H_S}, S) = 1 - \lambda + \lambda \log_2 |H_S|$.*

All proofs are deferred to Appendix B. In practice, of course the Generator may not find this uniform distribution but $|H_S| \geq 2^{v(D,S)}$ remains a lower bound for whatever $D$ is found.

## 3.2 LEARNING SOLVER

Next, consider an adaptive solver that can tailor its choice of $S \in \mathcal{S}$ by adjusting its parameters. Based on the theory of zero-sum games (Myerson, 2013), the game has a unique "value" that can be achieved by possibly different optimal "mixed strategies" which are probability distributions themselves. In the Problem-generation game, this is somewhat confusing as mixed strategies for the Generator are distributions over distributions of problems in $\Delta(\Delta(\mathcal{P}))$, and mixed strategies for the solver are distributions over solvers in $\Delta(\mathcal{S})$.

However, we next show that there is a single optimal strategy for the Generator (as long as each solver has 3 hard problems). Fortunately, this optimal Generator strategy is a "pure strategy", a single distribution $D^* \in \Delta(D)$ over problems rather than a distribution over distributions. Nonetheless, there may be multiple optimal strategies for the Learning Solver.

**Theorem 1.** *For any set $\mathcal{S}$ and $\lambda \in (0, 1]$, as long as each solver $S$ fails on at least $|H_S| \geq 3$ problems, there is a unique distribution $D^*$ that achieves the value of the zero-sum Problem-generation game.*

In particular, $D^*$ is simply the distribution that maximizes $J(D) = \min_{S \in \mathcal{S}} v(D, S)$. As mentioned, $\lambda = 0$ can admit multiple (degenerate) optimal solutions. Like Lemma 1, this Theorem is a theoretical statement as it is not clear that the Generator will find $D^*$.

---

**Algorithm 1:** The Troublemaker algorithm for finding parameters for the generator distribution. If a learning solver cannot accommodate weights, subsampling can be used to simulate weights.

---

**Input:** sample size $n$, number of steps $N$, step size $\eta$, a function that generates samples for any given $D_\theta$, a differentiable function that computes log-probabilities for $D_\theta$ and either: (a) fixed solver $S$ or, (b) Learning Solver that maps a weighted set of problems to solver $S$.

1 Choose $\theta_1 \in \Theta$
2 **for** $i \leftarrow 1$ **to** $N$ **do**
3      Create a problems sample $p_{i1}, p_{i2}, \ldots, p_{in}$ by independent draws from $D_{\theta_i}$
4      **if** $S$ *is a fixed solver* **then** $S_i \leftarrow S$
5      **else** feed problems $p_{i1}, \ldots p_{in}$ with weights $w_{ij} = 1 - \lambda - \lambda \log D_{\theta_i}(p_{ij})$ on problem $p_{ij}$ into the Learning Solver and let $S_i$ be its output. (In practice, the parameters of $S_{i-1}$ can be used as a warm start.)
6      $U_i \leftarrow$ the set of sample problems that $S_i$ failed to solve
7      Update by summing over problems that $S_i$ doesn't solve:
         $\theta_{i+1} = \theta_i + \frac{\eta}{n} \sum_{p \in U_i} \big(1 - 2\lambda - \lambda \log D_{\theta_i}(p)\big) \nabla_\theta \log D_{\theta_i}(p)$
8 **end**
9 **return** generator parameters $\theta_N$

---

### 3.3 TROUBLEMAKER ALGORITHM FOR SOLVING THE PROBLEM-GENERATION GAME

To find the distribution $D^*$, one may maximize $J(D) = \min_{S \in \mathcal{S}} v(D, S)$, which is concave in $D$:

**Observation 1.** *The function $J(D) = \min_{S \in \mathcal{S}} v(D, S)$ is concave in $D$.*

Hence, as a concave function, it can be theoretically maximized to within arbitrary precision, using the standard gradient-projection ascent, namely:

---

1 Choose $D_1$ to be uniform over $\mathcal{P}$
2 **for** $i \leftarrow 1$ **to** $N$ **do**
3      Find solver $S_i$ to minimize $v(D_i, S)$
4      Update $D_{i+1} = \Pi_{\Delta(\mathcal{P})}\big(D_i + \eta \nabla_D v(D_i, S_i)\big)$
5 **end**

---

Here we have used the fact that a gradient of a minimum of functions is the gradient of whichever function is minimal at that point. Note that minimizing $v(D_i, S)$ over $S \in \mathcal{S}$ is equivalent to finding the solver that solves the most problems where each problem $p$ has a non-negative *weight* $w_p = 1 - \lambda - \lambda \log D_i(p)$. Also, $\Pi_T$ denotes the projection on to the closest point in convex Euclidean set $T$, in this case the closest probability distribution. This gradient-projection method is known to converge at a rate that depends on $N$ and step size $\eta > 0$ (*e.g.* Zinkevich, 2003). However, this again is a theoretical algorithm assuming that one can represent an arbitrary distribution and can evaluate $v(S, D)$ over *all problems*, which is computationally intractable. In practice, one represents a Generator $D_\theta$ by parameters $\theta \in \mathbb{R}^n$ and optimizes them w.r.t. the samples drawn from the generator.

The TM algorithm, described in Algorithm 1, follows the alternating gradient ascent procedure above procedure except that, using a sampling approximation similar to the REINFORCE algorithm (Williams, 1992) except that the gradient calculation is changed since the reward depends on the probability of generation. It requires sampling from $D_\theta$ and computing gradients $\nabla_\theta \log D_\theta(p)$ of log-probabilities, as is standard for sequential neural generation procedures. One also needs to be able to compute a best solver in response to a distribution of problems, which a Learning Solver can also approximate from problems sampled from $D_\theta$. The analysis showing how the update in Step 4 is an unbiased estimate of the gradient of $J$ is given in Lemma 2 in Appendix B.

## 4 TROUBLEMAKER: GENERATING PROGRAMMING PUZZLES

Having derived a general form of the problem generation problem, we turn our attention to its instantiation in the Troublemaker framework. In this section, we first define a problem representation for puzzles used by our generator $D_\theta$, and then describe the architecture of the trainable generator.

```
bool equation ≔ term == term
float term ≔ 0.1  term + term  |  0.2  term * term
           |  0.05  term - term  |  0.05  term / term
           |  ...
           |  0.15  float(term == 0)  // evaluates to 1.0 if equality holds, 0.0 otherwise
           |  0.05  π  |  0.05  e  |  0.05  0.0  // constants
           |  0.3  x  // variable
```

Figure 2: An excerpt from our Probabilistic Context Free Grammar (PCFG) that defines a family of puzzles with floating-point solutions. Each production rule is annotated with a weight, automatically learned by the generator (see main text).

We assume a language of *programs* $L$ defined as a Context Free Grammar (CFG). Each abstract syntax tree (AST) $T \in L$ describes a programming puzzle problem $p_T : \mathcal{X} \to \{\top, \bot\}$, as defined in Section 3. When $T$ is clear from context, we write $p$ instead of $p_T$. For the purpose of this paper, we assume interesting problems are *hard* puzzles whose solution a solver cannot obtain within the given time budget *i.e.* $\texttt{time}(S, p_T) > B$. However, our framework is agnostic to this assumption and can be trivially modified to generate problems for different notions of "interesting."

To generate puzzles that involve complex reasoning patterns, the language $L$ needs to be expressive enough to represent a wide range of puzzles. For example, Figure 2 shows an excerpt from our grammar for generating floating-point puzzles (full definition in Appendix A). Due to its expressiveness, sampling programs from $L$ uniformly at random is not useful as it may generate overly simple problems or even unsolvable ones like $x^2 = -1$. We will first describe our core procedure for generating puzzles in $L$, and will then turn our attention to ensuring their complexity and solvability.

### 4.1 GENERATION MODEL

**Generation with Probabilistic Grammars**  The first generation strategy we consider is to employ a probabilistic grammar as shown in Fig. 2. As can be seen, depending on the solver, different rules can be encouraged to bias the production of problems. For instance, the weights in Fig. 2 display a higher preference for multiplication of terms – this lends itself conveniently to the generation of puzzles with higher-degree polynomials. Note that the probability of generating a program $p$ factorizes into its constituent production rules:

$$\Pr(p_T) = \prod_{r \in T} \Pr(r)$$

which constructively corresponds to a standard sampling procedure that builds the AST $T$ one production at a time by sampling a production to expand each nonterminal using its corresponding weight as an unnormalized probability. Further, as mentioned, we aim to produce *hard* problems, *i.e.* maximize $\log \texttt{time}(S, T)$ or, in general, any reward that fits into the framework described in Section 3. The weights $\theta$ of the probabilistic grammar can be learned using Algorithm 1, since sampling, computing the log-likelihood and its gradient are straightforward.

**Neural Guided Generation**  While this strategy allows us to tailor generation for a given solver, it only tunes global preferences (*i.e.* encourages certain rules *always*) as opposed to context-dependent changes (*i.e.* rule A is preferred conditioned on the current partial AST). To enable finer-grained control over the generation process, we propose a more versatile generation strategy that conditions prediction of a rule on the rules in a partial AST produced so far. The context-dependent conditioning model is parameterized as a trainable neural network, whose parameters guide the generation process.

Let $T_{<t}$ be a partial AST generated so far at the generation timestep $t$, assuming some fixed ordering of nonterminal expansions to generate the whole AST (we use the pre-order traversal). The *neural-guided generation strategy* parameterized the probability of the puzzle as

$$\Pr(p_T) = \prod_t \Pr(r_t \mid T_{<t})$$

where $r_t$ is the production rule expanded in the AST $T$ at the $t^{\text{th}}$ timestep. The probability $\Pr(r_t \mid T_{<t})$ is, in turn, parameterized as a neural network that takes as input a *partial tree embedding* $\phi(T_{<t})$ and outputs a distribution of probabilities for each syntactically valid production rule.

```
lhs == rhs → p**lhs == p**rhs      lhs == rhs → lhs**p == rhs**p
  lhs == rhs → p+lhs == p+rhs        lhs == rhs → p*lhs == p*rhs
```

Figure 3: Tree rewrite rules for `float` puzzles. Here $p$ is also $term$ from the grammar in Figure 2.

We consider two state-of-the-art approaches to parameterize the tree embedding model $\phi$. The first parameterization is a naïve baseline – encoding a traversal of the tree $T_{<t}$ with an LSTM (Hochreiter & Schmidhuber, 1997). While straightforward, it does not explicitly makes use of the syntax structure present in the AST. Motivated by recent research in program representation (Allamanis et al., 2017; Brockschmidt et al., 2018), we choose *graph neural networks* (GNNs) as a second tree embedding model. Specifically, the model employed is GNN-FiLM (Brockschmidt, 2019), a recent GNN architecture that allows for feature-wise linear modulations, achieving state-of-the-art performance on a range of graph tasks like node classification.

## 4.2 COMPLEX AND SOLVABLE PUZZLES

**Generating Solvable Programs**  In order to generate solvable puzzles, we use domain-specific knowledge to *convert* a puzzle sampled from $L$ to a solvable one. To continue with the `float`-puzzle example, we first select a random solution $x_0$. Then given an equation $a(x) == b(x)$ where both $a, b \in L_{term}$, we can convert it to a *solvable* puzzle at $x_0$ by modifying the program to be $a(x) = b(x) - k$ where $k = a(x_0) - b(x_0)$.

**Generating Complex Puzzles**  A standard strategy employed by experts for posing mathematical problems is *chaining* (Silver, 1994). Chaining expands on an existing problem (and solution) such that finding the solution for the modified problem requires solving the original problem as an intermediate step. For instance, in the case of our `float`-puzzle problems, exponentiating both sides of the equation is a good example of *chaining* – while preserving the *solvability*, it requires solving the original problem as an intermediate step (after taking log of both sides). In our setting of generating programming puzzles, we incorporate this problem posing strategy via tree-rewriting rules $R \colon L \to L$ that transform an existing AST to another AST in the language. For the example of `float`-puzzles, Fig. 3 provides a bank of tree-rewrite rules that can be used to produce complex solvable puzzles.

This approach is sufficiently general to apply to other domains such as `int`-puzzles or `set`-puzzles. For example, consider the sub-set sum problem – find set $\mathcal{B} \subseteq \mathcal{A}$ such that the sum of elements in $\mathcal{B}$ is a given integer $K$. If this `set`-puzzle is solvable, chaining can be used to produce more complex puzzles by replacing $\mathcal{A}$ with $\mathcal{A} \cup \mathcal{C}$ for some non-empty integer set $\mathcal{C}$. This transformation preserves solvability but can also make the puzzle harder, *e.g.* if the entire set $\mathcal{B} = \mathcal{A}$ is initially a solution.

## 4.3 TRAINABLE SOLVERS

Finally, while the Troublemaker framework as presented in Algorithm 1 is applicable for an arbitrary solver, it is particularly interesting for *trainable* solvers that can be improved with data produced by the generator. For them, the Troublemaker generation leads to an *adversarial* optimization process where the weights of the puzzle generator are updated to produce harder problems for the solver and similarly, the solver's weights are updated based on the newly generated *hard* problems. In this work, we study two variants of trainable solvers:

**Induction Solver (IS):** This neural solver directly outputs the solution $x$ of the puzzle $p_T$ given some embedding of the puzzle $\psi(T)$. In the case of structured outputs (like sets), the solver sequentially outputs elements along with a stop token to signal the end of generation.

**Synthesis Solver (SS):** This solver is similar to the neural generator – given a grammar $L$ along with a bank of useful constants, it constructs a solution $x$ as a *constant expression* in the grammar. The construction procedure is similar to puzzle generation – at each timestep $t$, it uses an encoding of the puzzle $\psi(T)$ and the current partial expression to select the next production rule to expand in the expression. Notably, while the generator emits the puzzle $p_T$ using an end-to-end differentiable LSTM, we found that synthesis of solutions $x$ is best parameterized as a discrete *search process* guided by the rule-prioritizing neural network as described above. This is similar to other state-of-the-art neuro-symbolic program synthesis techniques such as DeepCoder (Balog et al., 2016) and NGDS (Kalyan et al., 2018).

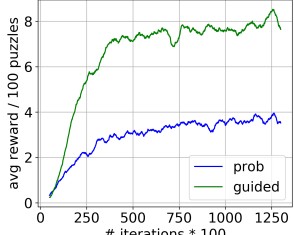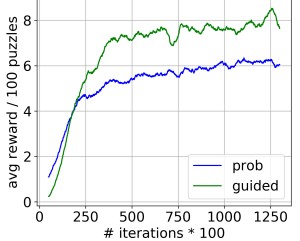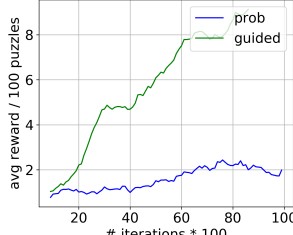

Figure 4: For the static solvers (Grid Solver, Enumerative and Sympy respectively), note that the reward achieved by both the probabilistic the neural-guided approach increases over time. Owing to its better expressivity and ability to model context, the guided approach latches on to the weakenesses of the solver faster than the probabilistic approach.

In both solvers, the puzzle embedding model $\psi(T)$ is a GNN-FiLM network, parameterized identically to the generator's embedding model (Section 4.1).

## 5 EXPERIMENTS

In this section, we present results for the TroubleMaker approach to generate hard problems for a given solver. We consider `float`, `int` and `intset` puzzles to demonstrate the flexibility of the proposed puzzle generation framework. We provide the full grammar and tree-rewrites in Appendix A.

**Puzzle Generators**  As discussed earlier, we study both probabilistic grammar based and neural-guided problem generators. Further, in the grammars used, we allow *copy* operations i.e. that can copy a node of type $\tau$ from the partial tree generated so far as opposed to generating a tree from $L_{NT}$ where $NT$ is of type $\tau$. The introduction of this operator helps the puzzle generator produce recursive structure (*e.g.* $10^{10} - x^x = 0$) – a desirable quality that often results in interesting reasoning patterns.

As discussed in Sec. 4.1, both generators are trained using Algorithm 1. Unless otherwise mentioned, all the solvers are capped to run within a time limit of 0.1s. Similarly, the size (# rules) of the puzzles is limited to a maximum of 20 and cannot exceed a depth of 10. Recall the definition of the reward function (Eq. 1) – ideally, a generator that maximizes this reward should have uniform support over "hard" puzzles (i.e. solver fails to solve within 0.1s) and at most be of size 20.

**Solvers**  We consider multiple solvers, both static and trainable. We now give their details below:

**Grid Search (GrS):**  This solver, as the name suggests, searches in the solution space, narrowing down to a satisfying solution based on comparisons. Unlike other solvers, a solution is passed off as correct if the equation holds within a specified tolerance. In our experiments, we use a fairly small tolerance of $10^{-32}$. Note that this solver is not a general purpose puzzle solver and is used only in the context of floating point and integer puzzles.

**Sympy** (https://www.sympy.org) is a Python library for symbolic mathematics and similar to GrS, it is used to only solve floating point and integer problems. Specifically, we use the `solve` function to evaluate the generator. Sometimes, despite arriving at the right solution, errors due to rounding the solution to floating point numbers causes the objective to be non-zero. Similar to GrS, we account for this by requiring the equation to hold within a specified tolerance – this check is performed only if the solver returns a solution.

**Enumerative Solver (ES):**  Given a grammar to generate the solution, this solver performs enumerative search to find a satisfying solution. For the sake of simplicity, we provide the solver with the same grammar used to generate a puzzle, albeit without the ability to produce variables. Apart from the constants already present in the grammar, the solver can also use constants extracted from the puzzle.

**Guided Solver (GS):**  This solver builds on top of the enumerative solver by using a neural network to guide the search process conditioned on the puzzle, as described in Section 4.3. By learning heuristics from data, this solver learns to accelerate enumerative search by pruning away large portions of the search space.

We also consider a simpler version of the guided solver where instead of a neural network, we have a pCFG like setting where a single weight determines the application of a rule. In both cases, the generator samples from the probability distribution produced on the set of rules (or rewrite rules).

**Training Details** As mentioned before, the generators are trained by maximizing for the reward in Eq. (1). The trainable solver *guides* the generation process exactly like the generator and is also trained via RE-INFORCE to maximize the number of solved puzzles. Additionally, the trainable solver is often warm started by using the traces of the enumerative solver. Both solvers and puzzle generators, when trainable, are optimized using Adam (Kingma & Ba, 2014) with a learning rate of $10^{-2}$. All the LSTM networks use a hidden size of 64, and all GNN-FiLM networks use 3 propagation steps and a node representation size of 64.

**Results** In terms of achieving higher reward values (see Fig. 4) and thus, producing more unsolvable problems for a static solver (see Fig. 5, the neural guided approach is better than the probabilistic grammar based generator. Both outperform randomly sampling puzzles from the grammar. Example generations with high reward are provided in Fig. 7.

In the case of learnable solvers, both the pCFG based and the guided solvers are trained for $N = 100$ iterations and each iteration $\sim 2000$

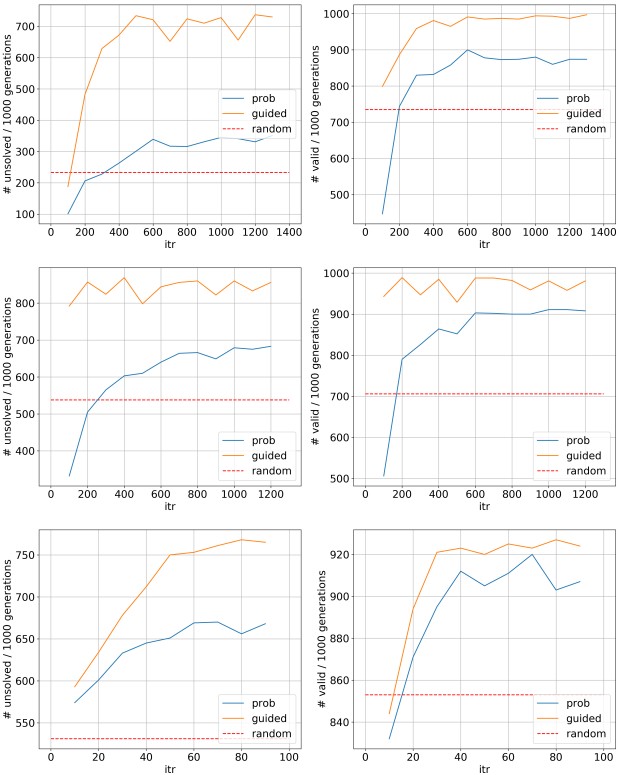

Figure 5: Number of unsolved and valid puzzles generated for each of the static solvers (Grid Solver, Enumerative, and Sympy respectively). While both the pCFG based and neural-guided generators find more "hard" puzzles over time, the guided solver saturates at a much higher value.

puzzles are sampled from the generator. From Figure 6, we see that the generator is always in a position to find more "hard" problems for the learning solver – likely because of building the guided solvers on top of the enumerative solvers. From qualitative examples presented in Figure 7, one can see that both the enumerative and the guided solvers have similar weaknesses (*e.g.* exponentiation) as appropriate inverse operations like square roots are not present in the grammar. Further, the solvers learned via TM are tuned to the distribution of problems from the corresponding generator – they solve only about 100 randomly sampled puzzles from the grammar (when evaluated at any iteration).

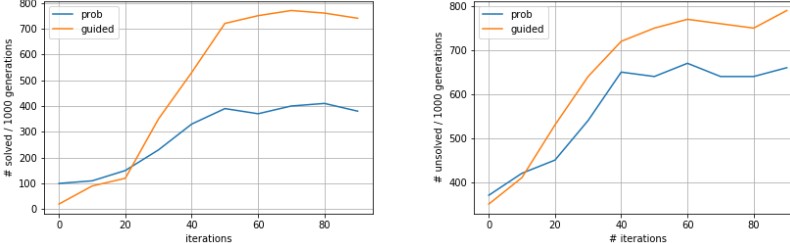

Figure 6: (**Left**) This figure shows the number of problems solved by the solver at each iteration – for every "iteration" of TM, both the generator and solver are updated. (**Right**) This figure shows the number of puzzles unsolved per 1000 generations. Critically, note that this and the previous plot are "offset" by an iteration *i.e.* the generator produces hard problems for the previously updated solver.

```
# grid solver:
(2. ** abs(math.sin(math.cos(math.log(abs(math.sin(math.sin(x)))))))) -
    (2. ** (1. + -0.4849144330195472)) == 0
(math.sin(math.sin(x)) ** math.pi) - ((x + 2.0833615479477763e-09) **
    math.pi) == 0

# enumerative solver:
(-x ** 2.) - ((x + 0.01653257550549251) ** 2.) == 0
(7. ** abs(math.cos(math.sin(math.sin(x))))) - (7. ** (x +
    3.8165676846602548)) == 0

# sympy solver:
(6. ** ((x ** abs(x)) ** x)) - (6. ** (9. + -8.004153605711155)) == 0
(((8. ** 4.) + 2.) + -8. ** 7.) - ((((8. ** 4.) + 2.) + -8.) ** ((
    float(math.log(math.sin(math.log(x))) == x)) + 7.0)) == 0

# learnable solver:
(5. + (math.log(x) / 2.)) - (5. + ((1. / abs(-x)) + -117.57345005874319))
    == 0 #iteration=10
((2. * x) ** math.pi) - (((math.cos(x * x)) + -39.63895930724463) **
    math.pi) == 0 #iteration=90
```

Figure 7: Qualitative examples of puzzles that achieve a high reward (sampled from top-100 of 1000 generations) for each static solver. In each of these cases, a neural-guided generator has been used to produce the puzzles. Each solver, has its specific weakness as can be seen from the examples – for example, excessive use of non-linear functions such as log, sin, and cos, make a problem hard for the grid solver. Similarly, simple exponentiation (here, via rewrite rules that simply exponentiate both sides) as the ability to compute the square root, etc. of a number is not provided in the grammar. Further, the generator games the sympy solver by frequently using exponentiation and absolute value. Note that the solver fails to solve some of the generator problems due to the time constraint, a fact exploited by the generator. As the learning solver is built on top of the enumerative solver, the generator in our setting overpowers the solver by producing puzzles with frequent exponentiations.

## 6 CONCLUSIONS

This work suggests a new type of problem called Programming Puzzles and a framework for generating a large set of hard problems that can both expose the weaknesses of existing solvers and which can be used in a GAN-like setup to train better solvers. Our system differs from prior work on data sets of problems, in that: (1) our puzzles are short, interpretable problems which may have multiple solutions but for which it is trivial to validate the accuracy of any solution, and (2) the puzzles are targeted specifically at any given solver. While it would be useful to have a ground-truth dataset of puzzles drawn from *e.g.* programming contests like ACM ICPC (see Appendix C), such a dataset is not strictly necessary for our worst-case approach.

One of the technical challenges that arises is that, in addition to optimizing for hardness, we also seek a large set of problems. Hence the reward of the puzzle generator takes into account the probability of each generated puzzle, which is uncommon in most GANs or reinforcement learning problems. Nonetheless, we show that the problem can be made well-defined in a differentiable pipeline, with a unique optimum solution that can be shown to converge. While our system is far from the final word on this problem, we believe these two ideas may initiate a line of work in which it is easier to make progress on the grand challenge of teaching computers to write complex programs.

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

(a) **bool** *equation* ≔ *term* == *term*
    **float** *term* ≔ *term* + *term* | *term* ⋆ *term*
             | *term* − *term* | *term* / *term*
             | *term⋆⋆term* | *−term*
             | **float**(*term* == 0) | **float**(*term* != 0)
             | cos(*term*) | log(*term*) | sin(*term*)
             | copy_float() *// copies a float node from current partial tree*
             | 0.0 | 1. | 2. | 3. | 4. | 5. | 6. | 7. | 8. | 9.
             | $\pi$ | $e$
             | $x$ *// variable*

(b) **bool** *equation* ≔ *int* **in** *term* | sum(*term*) == *int*
    **set** *term* ≔ ∅ | {*int_term*} | {*int_term, int_term*}
              | {*int_term, int_term, int_term*}
              | union(*term*, *term*) | intersection(*term*, *term*)
              | difference(*term*, *term*)
              | {e for e in *range(int_term, int_term)*}
    **int** *int_term* ≔ ... *// integer grammar*

Figure 8: Grammars that define programming puzzles with (a) float and (b) int-set solutions.

## A  GRAMMARS AND TREE-REWRITE RULES

We present the full grammars for the float and int-set domains used in this work in Figure 8. The integer rules are similar to the float domain except with additional functions factorial and modulo.

## B  THEORETICAL ANALYSIS AND PROOFS

*Proof of Lemma 1.* We can write the payoff as

$$v(D, S) = \sum_{p \in H_S} D(p)\big(\lambda(-\log D(p)) + 1 - \lambda\big) = (1 - \lambda)D(H_S) - \lambda \sum_{p \in H_S} D(p) \log D(p).$$

Clearly $(1 - \lambda)D(H_S) \leq 1 - \lambda$ is maximized if and only if all problems generated are hard for $S$. The second term $-\sum_{p \in H_S} D(p) \log D(p)$, is the entropy which is known to be maximized over the uniform distribution but is only summed over $H_S$. For example, if $\lambda = 1$ and $|H_S| = 1$, then the uniform distribution over $H_S$ would have a payoff of 0 because it has 0 entropy, and a distribution which put probability $1/e$ on this hard problem and probability $1 - 1/e$ on an easy problem would actually be optimal. However, whatever probability $D(H_S)$ is assigned to hard problems, it is straightforward to see optimality requires distributing this uniformly over $H_S$: by the strict concavity of the $-z \log z$ function there is a unique optimum and by symmetry it is uniform. Finally, since $-z \log z$ is increasing up to $z = 1/e > 1/3$, this quantity is only increasing in $D(H_S)$ as long as there are at least three hard problems, so the optimum is $D(H_S) = 1$ only over hard problems. □

To prove Theorem 1, a key technical observation is that $v(D, S) = \sum_{p \in H_S} D(p)\big(\lambda(-\log D(p)) + 1 - \lambda\big)$ is concave in $D$, which follows from the concavity of $-z \log z$ for $z \in [0, 1]$. Hence, as a zero-sum game with actions $D \in \Delta(\mathcal{P})$ and $S \in \mathcal{S}$, if the game were played with "mixed strategies" in which players could randomize amongst $D$ and $S$, it is a dominant strategy for the Generator never to randomize—instead it could achieve an expected payoff at least as large as any distribution over $D$'s by choosing the mean of the distributions. Hence, the "value" of this zero-sum game, given $\mathcal{P}, \mathcal{S}$ (and $\lambda$), is:

$$v(\mathcal{P}, \mathcal{S}) = \max_{D \in \Delta(\mathcal{P})} \min_{S \in \mathcal{S}} v(D, S).$$

In fact, this optimization has a unique optimal distribution $D$ (which is not true of zero-sum games in general, and there may be multiple optimal mixed strategies for the Solver):

*Proof of Theorem 1.* The minimax theorem (Neumann, 1928) implies that,

$$v(\mathcal{P}, \mathcal{S}) = \max_{D \in \Delta(\mathcal{P})} \min_{S \in \mathcal{S}} v(D, S) = \min_{\mu \in \Delta(\mathcal{S})} \max_{D \in \Delta(\mathcal{P})} \mathbb{E}_{S \sim \mu}[v(D, S)],$$

where we have simplified again by the concavity of $v$ in $D$ in not consider distributions over $D$'s. In particular, let $\mu^*$ be any minimizer of the right-hand-side above. Any maximum $D^*$ of $\min_{S \in \mathcal{S}} v(D, S)$ must also maximize $\mathbb{E}_{S \sim \mu^*}[v(D, S)]$, which we denote by,

$$v(D, \mu^*) = \mathbb{E}_{S \sim \mu^*}[v(D, S)] = \sum_p \mu^*(\{p \mid p \in H_S\})D(p)\big(\lambda(-\log_2 D(p)) + 1 - \lambda\big).$$

It remains to argue that there is a unique $D^*$ which maximizes the above quantity. Note that $-\log_2 D(p) \geq 0$ so the quantity in the right-hand parentheses is non-negative. Also note that $-D(p) \log D(p)$ is increasing for $0 \leq D(p) < 1/e$.

Let $h_p = \mu^*(\{p \mid p \in H_S\})$ be the hardness of $p$, the probability a random solver from $\mu$ doesn't solve $p$. We do this by arguing, first, that any maximal $D^*$ would assign $D(p) = 0$ to any "easy" puzzle $p$ such that $h_p = 0$. To see this, note that since every solver has three puzzles that are hard for it, there must be at least three puzzles for which $h_p > 0$ and at least one of these call it $p'$ must have $D(p') \leq 1/3 < 1/e$ and $h_{p'} > 0$. Hence, $v(D, \mu^*)$ could be increased by taking an amount of probability $D(p)$ assigned to a $p$ with $h_p = 0$ and moving it to $p'$.

Therefore $D$ distributes its probability only over $p$ with $h_p > 0$. Since $-D(p) \log D(p)$ is strictly concave and $D(p)$ is trivially concave (as a linear function), the quantity of interest,

$$\sum_{p:h_p>0} h_p D(p)\big(\lambda(-\log_2 D(p)) + 1 - \lambda\big)$$

is strictly concave in $D$ and hence has a unique maximum. To see why you need some lower-bound on the number of hard puzzles, note that if only one puzzle $p'$ was hard for all solvers $\forall S\ H_S = \{p'\}$, $\lambda = 1$, and all other puzzles were easy, then the optimum solution would not put all $D$ on the hard puzzle. Instead it would be optimal to assign $D(p') = 1/e$ and spread the remaining $1 - 1/e$ fraction of probability arbitrarily among the puzzles where $h_p = 0$. $\qquad\square$

We now discuss how Step 4 of the TM algorithm (see Figure 1) yields an unbiased estimate of the gradient $\nabla_\theta J(D_\theta)$ from samples.

**Lemma 2.** *For any Generator $D_\theta$ and solver $S$, over random iid samples $p_1, \ldots, p_n$ from $D$,*

$$\nabla_\theta v(D_\theta, S) = \mathbb{E}_{p_1, \ldots, p_n \sim D}\left[\frac{1}{n} \sum_{j:p_j \in H_S} \big(1 - 2\lambda - \lambda \log D_\theta(p_j)\big)\nabla_\theta \log D_\theta(p_j)\right].$$

*Again, $p_j \in H_S$ means that solver $S$ did not solve problem $p_j$.*

*Proof.* We first compute the gradient of $J(D_\theta)$ with respect to $\theta$:

$$\begin{aligned}
\nabla_\theta J(D_\theta) &= \nabla_\theta \mathbb{E}_{p \sim D_\theta}[r(p, S, D_\theta)] \\
&= \sum_p \nabla_\theta\big(D_\theta(p) r(p, S, D_\theta)\big) \\
&= \sum_p r(p, S, D_\theta)\nabla_\theta D_\theta(p) + D_\theta(p)\nabla_\theta r(p, S, D_\theta) \\
&= \sum_p r(p, S, D_\theta)D_\theta(p)\nabla_\theta \log D_\theta(p) + D_\theta(p)\nabla_\theta r(p, S, D_\theta) \\
&= \mathbb{E}_{p \sim D_\theta}\big[r(p, S, D_\theta)\nabla_\theta \log D_\theta(p) + \nabla_\theta r(p, S, D_\theta)\big]
\end{aligned}$$

In the second-to-last line above, we have used the standard "REINFORCE trick" writing $\nabla f(\theta) = f(\theta)\nabla \log f(\theta)$ to make it writable as an expectation. Also note that above applies to an arbitrary reward whose gradients may be computed by automatic differentiation, so the approach generalizes to other reward functions.

Now, for our specific reward function defined in Eq. (1), $\nabla_\theta r(p, S, D_\theta) = -\mathbf{1}_{p \in H_S}\lambda\nabla_\theta \log D_\theta(p)$, where $\mathbf{1}_P$ for proposition $P$ is the indicator function that is 1 if $P$ holds and 0 otherwise. Using this notation, $r(p, S, D_\theta) = \mathbf{1}_{p \in H_S}(1 - \lambda - \lambda \log D_\theta(p))$. Hence, we have,

$$\begin{aligned}
\nabla_\theta J(D_\theta) &= \mathbb{E}_{p \sim D_\theta}\big[\mathbf{1}_{p \in H_S}(1 - \lambda - \lambda \log D_\theta(p))\nabla_\theta \log D_\theta(p) - \mathbf{1}_{p \in H_S}\lambda\nabla_\theta \log D_\theta(p)\big] \\
&= \mathbb{E}_{p \sim D_\theta}\big[\mathbf{1}_{p \in H_S}(1 - 2\lambda - \lambda \log D_\theta(p))\nabla_\theta \log D_\theta(p)\big]
\end{aligned}$$

```python
def score_azulejos(s: List[int], t: List[int],
                   p1=[3, 2, 1, 2], h1=[2, 3, 4, 3],
                   p2=[2, 1, 2, 1], h2=[2, 2, 1, 3]):
    n = len(p1)
    return (
        sorted(s) == sorted(t) == list(range(n))
        and
        all(h1[s[i]]>h2[t[i]] for i in range(n))
        and
        all(p1[s[i]]<=p1[s[i+1]] for i in range(n-1))
        and
        all(p2[t[i]]<=p2[t[i+1]] for i in range(n-1))
    )
```

Figure 9: ICPC Problem 2019A in English (left) and programming puzzle (right). In the PSAT, `p1`, `p2` are arrays of $n$ prices, `h1`, `h2` are arrays of $n$ heights. The goal is to find `s`, `t`, required to be lists of integers in the function specification. They are further tested to be permutations by checking if sorting them yields `list(range(n))`, the list of numbers from 1 to `n`.

By summing over those samples for which $p_j \in H_S$, the quantity in the lemma is seen to be equal to the above in expectation. □

## C  ICPC PROBLEM

As a somewhat complicated but illuminating example, consider the first problem of the 2019 International Collegiate Programming Contest (ICPC). At its core, the goal is, given input two matrices $P, H \in \mathbb{R}^{2 \times n}$, to find a pair of permutations $\sigma, \tau$ on $\{1, 2, \ldots, n\}$ such that,

$$h_{1\sigma_i} < h_{2\tau_i} \text{ and } p_{1\sigma_i} \leq p_{1\sigma_{i+1}} \text{ and } p_{2\tau_i} \leq p_{2\tau_{i+1}} \text{ for all } i$$

The permutations $\sigma$ and $\tau$ are to be applied to the two rows of $P$, which correspond to prices. When permuted, the prices are to be non-decreasing. If there are duplicate values in $P$, this constraint leaves flexibility in $\sigma$ and $\tau$ which must be chosen so as to that when heights, encoded in $H$, are also permuted by $\sigma$ and $\tau$, the first row is greater than the second coordinate-wise. Figure 9 shows the problem in English and our PSAT version.

Note that there are a few important differences:

**Input parsing**. First, the problem like many requires reading and parsing the data from a file rather than receiving it as lists of integers. However, much prior work has successfully shown how to automatically parse such data (see the survey by Gulwani et al., 2017), so we focus on the algorithmic questions.

**Solvability**. Second, the problem posed in the contest asks users to print "impossible" if the problem has no solution. For such problems, the corresponding puzzle is unsolvable – there simply aren't any valid input which makes it return True. If one wanted to have an objective question where one could verify that an answer of "impossible" was correct, the puzzle would need some sort of proof that the problem is unsolvable which is much more difficult. However, the uncertainty of knowing whether an unsolved puzzle is solvable is not critical for optimization. For example, imagine a solver takes a test with 1,000 problems and knows with certainty that one solved 300 of them and didn't solve the remaining 700. Now, suppose changing a parameter of the system makes it so that it solves an additional 20 problems. This increase in objective is desirable, and even though a humans may be frustrated and even slower to solve a problem without knowing whether or not it is solvable, a computer algorithm will experience no such frustration or slowdown.

**Solving instances**. For people, there is a difference between submitting a program that solves a given problem with varied inputs and just submitting solutions for a number of given inputs – the latter may be easier as they may identify peculiarities in the inputs which make them easier to solve (or they may even solve them by hand). However, for computers this distinction is not crucial. Given a program

| `float`-puzzles | | | | |
|---|---|---|---|---|
| Puzzle Generators | Solvers | | | |
| | Enumerative Solver | Guided Solver | Grid Solver | Sympy |
| Random | 1.21 | 0.92 | 1.5 | 0.05 |
| Prob | 2.47 | 1.95 | 2.22 | 1.02 |
| Guided | 2.78 | 1.99 | 2.34 | 1.45 |
| `int`-puzzles | | | | |
| Random | 3.06 | 2.93 | 3.78 | 4.21 |
| Prob | 3.22 | 2.95 | 3.82 | 4.33 |
| Guided | 3.45 | 3.15 | 3.85 | 4.54 |
| `int-set`-puzzles | | | | |
| Random | 4.12 | 3.83 | - | - |
| Prob | 4.33 | 4.02 | - | - |
| Guided | 4.51 | 4.19 | - | - |

Table 1: As can be seen from the table, both generators – probabilistic grammar based (Prob) and neural guided (Guided) lead to improvements (over randomly sampled puzzles) in the **time** required for the solvers, with the guided solver showing the highest gains. Note that randomly sampled puzzles can sometimes be unsolvable – in which case the solver just times out. Further, all values are measured in seconds and solvers have a time bound $B = 5s$.

that is only capable of solving individual instances, that same program can itself be submitted as a solution to the general problem. This distinction is similar to a classification problem in which a binary classifier must be returned versus one in which unlabeled test data is provided and the labels alone must be submitted. Similar techniques are often used for computers to solve these two closely-related problems if the test set is sufficiently large.

## D EXPERIMENTS: MAXIMIZING TIME

Table 1 shows the results for different generators when optimized for `logtime` instead of the reward function discussed in Equation (1).

