# OpenReview forum: "ADAPTIVE GENERATION OF PROGRAMMING PUZZLES"
_ICLR.cc/2020/Conference — Reject_

### Official Review · AnonReviewer3 · 2019-10-24
**Official Blind Review #3**

**Rating:** 8

**Review:**

This paper proposes a method for generating hard puzzles with a trainable puzzle solver. This is an interesting and important problem which sits at the intersection of symbolic and deep learning based AI. The approach is largely GAN inspired, where the neural solver takes the role of a discriminator, and the generator is trained with REINFORCE instead of plain gradient descend.

Although I'm not an expert in this area, I have found this paper well written and easy to follow. The problem is well motivated, and the approach is sensible. As this is a novel problem, the paper also defines their own metric, namely the average time taken to solve the puzzle by given solvers, and the diversity of generated puzzles. It is nice to see that the generator indeed learns to generate puzzles that are significantly harder than random counterparts, while maintaining reasonable diversity. Although I think these are convincing results, my question to the authors is: have you tried or considered other ways of evaluating the generated puzzles? E.g., if you train the guided search solver on the generated puzzles and evaluate it on a random set of puzzles, would you see an improvement? I think this would be interesting to see, which can serve as an alternative evaluation metric.

My other comments are regarding the experiment section:
1. It would be useful to provide references to the solvers used, both in the adversarial training phase and the evaluation phase, if there is any.
2. More details of the training process would also be valuable. E.g., the training time and stability, common failure modes if any.

Minors:
1. Figure f3 should be s.count("A")==1000 and s.count("AA")==0
2. First sentence under Fig 1, one is give -> one is given
3. Figure 5, f2: 2**(x**2)) == 16 -> 2**(x**2) == 16

**Experience Assessment:**

I do not know much about this area.

**Review Assessment: Checking Correctness Of Derivations And Theory:**

I assessed the sensibility of the derivations and theory.

**Review Assessment: Checking Correctness Of Experiments:**

I assessed the sensibility of the experiments.

**Review Assessment: Thoroughness In Paper Reading:**

I read the paper at least twice and used my best judgement in assessing the paper.

---

> ### Author Response · Authors · 2019-11-12
> **Evaluation of Guided Generator**
>
> Thank you for the great review and improvement suggestions! We are currently working on the new revision, and will incorporate your suggestions and other improvements into it in a few days. This will also include more details on the training process, experiments, and references to the used solvers embedded in Related Work.
>
> Q: If you train the guided search solver on the generated puzzles and evaluate it on a random set of puzzles, would you see an improvement? This is indeed an interesting question, and would be a useful additional metric for evaluating the generator’s quality. At the moment, the focus of this work is generating a hard and diverse dataset of puzzles for *assessing* artificial solver algorithms. That said, generating training curricula would be a reasonable extension of the same framework and a good focus for future work.
>
> As an additional evaluation question, we will be adding results on the trained solver’s performance on randomly sampled problems from the grammar to the new paper revision.

---

### Official Review · AnonReviewer2 · 2019-10-27
**Official Blind Review #2**

**Rating:** 3

**Review:**

This paper proposes a trainable 'puzzle' program synthesizer that outputs a program f with a specific syntax. These 'puzzles' are structured as boolean programs, and a program solver solves the puzzle by finding an input x such that f(x) = True.  The authors motivate this task by making a case that puzzles of this sort are a good domain for teaching computers how to program.

The paper is fairly clear overall. There is some repetition in the early parts, so this could be restructured a bit, but these are minor points. A more significant restructuring however, is that this work would benefit from the related work being present the beginning of the work. Since this work is so similar in many ways to previous work I think the overall clarity of the paper would be improved, and the contributions clearer, if the work was better situated with respect to related work.

The experiments demonstrate that the trainable puzzle generator is able to produce harder (i.e. takes longer time to solve) puzzles than a random or probabilistic generator of the same grammar. While this does show that the program generator is learning something useful, these results are insufficient to show the utility of this approach in any real context. It seems the most interesting solver to assess is a trainable solver. Yet only 1 of the 4  solvers they assess is trainable. I know the authors make a point that they are not putting forward any new solver algorithms. That ok, however, taking existing trainable solvers and assessing how they perform with this guided puzzle generation vs. some other puzzle generation approach is a critical empirical study. Furthermore, it would be helpful to have more discussion of the baseline methods of generating puzzles. When the trainable puzzle solver was originally proposed, how was it trained? Where did the  data come from? How does that compare to this approach. I am not very familiar with this literature, and I imagine this paper would be of interest to folks outside the program synthesis space, so it would be very helpful to better explain this (also we note about related work). There are several additional empirical analysis that could be added to improve this work. For example, for the trainable solver, a plot of (training time) vs (time to solve puzzle) would be interesting. Beyond looking at training time, does a solver trained with the guided puzzle generator end up being a 'better' solver in some way? Are the resulting puzzles harder but still being solved? Is there a way of quantifying the 'hardness' of a puzzle? Perhaps a proxy like size? Then it would be cool to plot (training time) vs (approx puzzle hardness) to demonstrate that . the puzzle generator is really developing a reasonable curriculum.

Finally, there is a bunch of related work that I think is missing. Again, I'm not super familiar with this work, but I think there is a lot of curriculum learning stuff within RL that seems super relevant. Of particular relevance is the Alice/Bob framework from "Intrinsic motivation and automatic curricula via asymmetric self-play" seems very similar to the work at hand.  Something that is interesting in the Alice/Bob framework that could be transferred over here is the notion of the generator wanting to make a puzzle hard, but not too hard, i.e. make it just outside the solvers current capabilities.

Overall my assessment is that this paper doesn't quite meet the standard for ICLR. My two major critiques are (1) the related work is seriously lacking making it difficult to situate this work in a broader context. The authors also seem to miss the entire curricular learning literature. (2) The empirical evaluations are lacking. In particular, more thorough analysis of how the generator is behaving, the type of curriculum it learns, and the resulting impact this has on a trainable solver all are missing. Furthermore, more focus on trainable solvers would improve this work.

I'm not an expert in this area so it is possible I misjudged the significance of this work. I'm certainly open to revising my assessment if the authors are able to address (2) in a meaningful way.



**Experience Assessment:**

I do not know much about this area.

**Review Assessment: Checking Correctness Of Derivations And Theory:**

N/A

**Review Assessment: Checking Correctness Of Experiments:**

I assessed the sensibility of the experiments.

**Review Assessment: Thoroughness In Paper Reading:**

I read the paper at least twice and used my best judgement in assessing the paper.

---

> ### Author Response · Authors · 2019-11-12
> **Related work (curriculum learning) and other empirical evaluations**
>
> Thank you for the detailed review and great suggestions! We are currently working on the new revision, and will incorporate your suggestions and other improvements into it in a few days.
>
> Relation to Curriculum Learning: The contribution of our work is two-fold -- we introduce programming puzzles as a useful class of reasoning problems and then propose “troublemaker” algorithm to generate hard puzzles for a given solver. We agree that generating training curricula is an interesting application of our (or similar) framework; however, it is not the focus of this work. The goal of the troublemaker algorithm in this work is to generate a set of hard puzzles that shed light on the weaknesses of existing solvers. When the solver is trainable, we propose an adaptive version of the troublemaker algorithm that improves the puzzle generator iteratively.
>
> That said, we agree that the curriculum learning literature is highly relevant, even if this work does not explore curriculum learning per se. We are including an overview of the relevant ideas into Related Work in our next revision (and will move it earlier in the PDF, as you rightly suggest).
>
> Evaluation of Trainable Solver. As you rightly point out, we do not focus on developing new solvers, trainable or otherwise. In this work, we evaluate the trainable solver against two generation mechanisms -- 1) Probabilistic context-free grammar based and 2) Neural Guided. In both cases, the weights (rule weights for pCFG and neural network parameters in the case of guided generators) are updated based on the solver performance. While table 1 provides a comparison, we were unable to add more empirical analysis due to time and space constraints.
>
> On the float grammar, we find that the guided generator produces about 200/1000 unsolvable problems (same when 1000 puzzles are randomly sampled from the grammar) and after about 1000 iterations saturates to produce ~700/1000 unsolvable problems. On the other hand, the pCFG based generator saturates at only ~350/1000 unsolvable problems after 500 iterations; nearly half the number of unsolvable problems produced by the guided generator on average. We will add detailed analysis/plots to the next updated draft.
>
> Further, we will add a more detailed analysis of the learning solver in the next version. We will add a comparison of the solver’s performance at various stages of training and compare the effect of training time vs. solver performance (# puzzles solved). Additionally, we will evaluate a trainable solver and evaluate its performance on randomly sampled puzzles from our grammar.
>
> We *may* be able to add another flavor of a trainable solver in time for the discussion period, to show how the Troublemaker generation influences different kinds of training processes (as you rightly point out).
>
> Q. How is the initial trainable solver trained?
> The solver must be pre-trained on puzzles from the same grammar/distribution we consider in this paper, thus it cannot be lifted from some prior work on other datasets (e.g. Saxton et al). For the first iteration of the Troublemaker algorithm, we sample puzzles from the grammar uniformly at random and use the traces of an enumerative solver to train it. Following this phase, it can be directly trained via REINFORCE as the puzzles we generate are “checkable” i.e. the validity of a solution can be verified. As adaptive training progresses, the solver is continuously retrained on the puzzles sampled from the Troublemaker generator (as well as all the previously generated puzzles). We will clarify this better in the paper.
>
> Q. Is there a way to quantify the hardness of a puzzle? Size? This is an interesting question that is non-trivial to answer. For the purpose of this work, we define hardness based on the time taken to solve a puzzle. For example, when we state that our algorithm finds “hard” problems for a given solver we mean that the solver is unable to solve the problem within some time limit.
>
> From preliminary experiments, we find that problem size is not a good measure of hardness i.e. the number of unsolvable problems (for a given solver) does not increase with the size of the problems generated. We will update the draft with detailed analysis.
>
> Q. Training time vs. puzzle hardness. We find that both the pCFG based and neural guided generators tend to produce rapidly increase in the number of unsolvable produced before saturating at a certain value. We find that the guided generator nearly produces twice the number of unsolvable problems as the pCFG based generator (and ~4 times that of random sampling) We will add these plots in the next revision.

---

### Official Review · AnonReviewer4 · 2019-11-08
**Official Blind Review #4**

**Rating:** 3

**Review:**

In this paper, the authors propose a new class of programs they call programming puzzles. The authors argue that this class of programs is ideal for helping learn AI systems to reason. The second contribution of the paper is an adaptive method of puzzle generation inspired by GAN-like generation that can generate a diverse and difficult set of programs. The paper shows that the generated puzzles are reasonably difficult to solve (using the time to solve as a measure of difficulty) and reasonably diverse.

I found the paper well-written and easy to understand. The methodology to generate programs is convincing. I am not sure time to solve is the best way to measure the complexity of the program, but it seems a reasonable proxy. Did the authors study if the program length is correlated to the time to execute? If the correlation holds, then can complex programs not be created by simply having a bias towards longer programs? That would be a strong baseline to compare against.

I may have missed something, but I understand that only the Guided Solver is trainable. If that is the case, then why do we see increase in solving time for other solvers (Table 1). In only the case of the Guided solver can the generator adaptively increase the complexity of the programs.

Overall, I feel that the paper puts forth an interesting class of programs. But there are some gaps in the evaluation and the baselines. I am also not sure how this class of programs can help advance artificial reasoning.

Feedback:
- The authors should provide more references to the solvers used in Section 5.
- The paper ends abruptly. A summary/conclusion would be useful.

I am not an expert in this area, and I am willing to revise my recommendation if the authors can address these issues.


**Experience Assessment:**

I do not know much about this area.

**Review Assessment: Checking Correctness Of Derivations And Theory:**

N/A

**Review Assessment: Checking Correctness Of Experiments:**

I assessed the sensibility of the experiments.

**Review Assessment: Thoroughness In Paper Reading:**

I read the paper at least twice and used my best judgement in assessing the paper.

---

> ### Author Response · Authors · 2019-11-12
> **Advantages of programming puzzles, problem size vs. hardness and non-trainable solvers**
>
> Thank you for your great review! We are currently working on the new revision, and will incorporate your suggestions and other improvements into it in a few days.
>
> Q: How does this class of programs help advance artificial reasoning? Programming puzzles, in contrast to many other tasks used to evaluate artificial intelligence at the time, have three distinct features:
> They are objective: an answer is easy to validate and score automatically.
> They require abstract reasoning but not real-world knowledge, NLP, or spatio-temporal biases.
> This class of reasoning problems occupy a sweet spot in complexity: often, humans can solve programming puzzles easily, but current reasoning systems fail or take a long time to solve.. Thus, it’s a great next milestone for advancing the capabilities of artificial reasoning.
>
> Q: Is the program length correlated with time to execute? No, there is no correlation with the program size and the execution time. Further, from preliminary experiments, we don’t find that the number of problems unsolved (by the solver) does not increase with problem size. (please also see our response to Reviewer 2).
>
> The baseline that only generates big problems is an excellent suggestion, thank you! We will be running this evaluation over the next few days.
>
> Q: Why do non-trainable solvers also increase their processing time over the course of training? Because the Troublemaker generator discovers their weaknesses (i.e. patterns in the puzzle construction that the solver is not well-equipped to handle) over time. This is the key feature of our adaptive algorithm. Non-trainable solvers, by construction, are designed to solve a class of puzzles well (and often all other classes poorly or not at all). The fact that Troublemaker automatically discovers this class over time serves as a validation of its core capabilities.
>
> We will also add detailed discussions on the construction of the solvers to the appendix.

---

### Author Response · Authors · 2019-11-15
**Summary of changes made during rebuttal period**

Thank you for your feedback --- it has helped us improve the paper immensely! In particular, we have uploaded a paper with several changes including:
- We have expanded on the related work section situating our contributions; and moved it earlier in the paper.
- We have added a game-theoretic analysis of the Problem-generation game, putting our algorithm on more solid foundations.
- We have added an analysis of the behavior of the generator and the learnable solver as a function of training time.
- We have added another learnable solver that updates weights for each rule, similar to the pCFG based solver proposed in the paper (and corresponding results).
- Additionally, we performed an analysis of the trainable solver on randomly generated problems. We notice that it does not fare well (worse than enumerative solver) — we note this observation in the results section — somewhat expected as the solver is tuned towards puzzles produced by the generator.
- We have added a short conclusion section.

---

### Decision · Program_Chairs · 2019-12-19

**Decision:**

Reject

**Comment:**

The authors introducing programming puzzles as a way to help AI systems learn about reasoning. The authors then propose a GAN-like generation algorithm to generate diverse and difficult puzzles.

This is a very novel problem and the authors have made an interesting submission. However, at least 2 reviewers have raised severe concerns about the work. In particular, the relation to existing work as pointed by R2 was not very clear. Further, the paper was also lacking a strong empirical evaluation of the proposed ideas.  The authors did agree with most of the comments of the reviewers and made changes wherever possible. However, some changes have been pushed to future work or are not feasible right now.

Based on the above observations, I recommended that the paper cannot be accepted now. The paper has a lot of potential and I would strongly encourage a revised submission addressing the questions/suggestions made by the reviewers.